# Gender Differences in Psychological Symptoms and Quality of Life in Patients with Inflammatory Bowel Disease in China: A Multicenter Study

**DOI:** 10.3390/jcm12051791

**Published:** 2023-02-23

**Authors:** Chuan Liu, Jixiang Zhang, Min Chen, Ping An, Jiankang Xiang, Rong Yu, Suqi Zeng, Shuchun Wei, Beiying Deng, Zhongchun Liu, Changqing Jiang, Jie Shi, Kaichun Wu, Weiguo Dong

**Affiliations:** 1Department of Gastroenterology, Renmin Hospital of Wuhan University, Wuhan 430060, China; 2Department of Gastroenterology, Xijing Hospital, Air Force Medical University, Xi’an 710032, China; 3Department of Psychiatry, Renmin Hospital of Wuhan University, Wuhan 430060, China; 4Department of Clinical Psychology, Beijing Anding Hospital, Capital Medical University, Beijing 100088, China; 5Department of Medical Psychology, Chinese People’s Liberation Army Rocket Army Characteristic Medical Center, Beijing 100088, China

**Keywords:** inflammatory bowel disease, gender differences, psychological symptoms, sleep quality, quality of life, nomogram, survey

## Abstract

Objective: To explore the gender differences in the psychological symptoms, sleep quality, and quality of life of patients with inflammatory bowel disease (IBD). Methods: A unified questionnaire was developed to collect clinical data on the psychology and quality of life of IBD patients from 42 hospitals in 22 provinces in China from September 2021 to May 2022. The general clinical characteristics, psychological symptoms, sleep quality, and quality of life of IBD patients of different genders were analyzed via a descriptive statistical analysis. A multivariate logistic regression analysis was conducted, and independent influencing factors were screened to construct a nomogram to predict the quality of life. The consistency index (C-index), receiver operating characteristic (ROC) curve, area under the ROC curve (AUC), and calibration curve were used to evaluate the discrimination and accuracy of the nomogram model. Decision curve analysis (DCA) was used to evaluate the clinical utility. Results: A total of 2478 IBD patients (1371 patients with ulcerative colitis (UC) and 1107 patients with Crohn’s disease (CD)) were investigated, including 1547 males (62.4%) and 931 females (37.6%). The proportion of anxiety in females was significantly higher than in males (IBD: 30.5% vs. 22.4%, *p* < 0.001; UC: 32.4% vs. 25.1%, *p* = 0.003; CD: 26.8% vs. 19.9%, *p* = 0.013), and there were differences in the severity of anxiety between the genders (IBD: *p* < 0.001; UC: *p* < 0.001; CD: *p* = 0.050). The proportion of depression in females was higher than in males (IBD: 33.1% vs. 27.7%, *p* = 0.005; UC: 34.4% vs. 28.9%, *p* = 0.031; CD: 30.6% vs. 26.6%, *p* = 0.184), and there were differences in the severity of depression between the genders (IBD: *p* = 0.004; UC: *p* = 0.022; CD: *p* = 0.312). The proportion suffering from sleep disturbances among females was slightly higher than among males (IBD: 63.2% vs. 58.4%, *p* = 0.018; UC: 63.4% vs. 58.1%, *p* = 0.047; CD: 62.7% vs. 58.6%, *p* = 0.210), and the proportion of females with a poor quality of life was higher than that of males (IBD: 41.8% vs. 35.2%, *p* = 0.001; UC: 45.1% vs. 39.8%, *p* = 0.049; CD: 35.4% vs. 30.8%, *p* = 0.141). The AUC values of the female and male nomogram prediction models for predicting poor quality of life were 0.770 (95% CI: 0.7391–0.7998) and 0.771 (95% CI: 0.7466–0.7952), respectively. The calibration diagrams of the two models showed that the calibration curves fitted well with the ideal curve, and the DCA that showed nomogram models could bring clinical benefits. Conclusions: There were significant gender differences in the psychological symptoms, sleep quality, and quality of life of IBD patients, suggesting that females need more psychological support. In addition, a nomogram model with high accuracy and performance was constructed to predict the quality of life of IBD patients of different genders, which is helpful for the timely clinical formulation of personalized intervention plans that can improve the prognosis of patients and save medical costs.

## 1. Introduction

Inflammatory bowel disease (IBD), which includes ulcerative colitis (UC) and Crohn’s disease (CD), is a systemic, autoimmune, recurrent, remitting gastrointestinal disease, and its peak incidence is generally between 20 and 30 years old [1]. It has a worldwide trend of increasing incidence year by year [1]. Epidemiological studies [2,3,4,5,6,7] have confirmed that there is a significant gender difference in the incidence of IBD and that such variation shows obvious regional differences. In European and American countries such as the United States [2], Canada [3], and Denmark [4], the incidence in women is higher than in men. In Asian countries, such as Korea [5], India [6], and China [7], the incidence is higher in men than in women. At the same time, a large-sample population study [8] showed that the prevalence of psychological distress with impairment was significantly higher in IBD than non-IBD adults. A global meta-analysis of 77 studies, including 30,118 patients, showed that the prevalence of anxiety symptoms in IBD patients was 32.1% and the prevalence of depression symptoms in IBD patients was 25.2% [9]. As for the relationship between the psychological status of IBD patients and gender differences, some studies have pointed out that female IBD patients are more likely to suffer from psychological problems such as anxiety and depression than male patients [9,10]. For example, the prevalence of comorbid anxiety and depression in female IBD patients is 33.8% and 21.2% compared to 22.8% and 16.2% in male IBD patients, respectively [9].

Screening people with IBD for psychological disorders has recently been shown to have mental health benefits [11]. Patient questionnaires are readily available and can be used to help diagnose anxiety and depression [12]. Attention has increasingly been paid to the psychological problems of IBD patients. However, the gender difference in IBD patients has only been paid attention to in recent years, and there are few relevant studies. In particular, there are no studies that specifically explore the gender differences in the clinical and mental health of IBD patients, which is of great significance for individualized diagnosis and treatment because precision medicine is the future trend of IBD diagnosis and treatment [13]. Therefore, this study conducted a cross-sectional survey with a large sample of Chinese patients from multiple centers in order to ascertain the current status of gender differences in the general clinical characteristics, psychological symptoms, sleep quality, and quality of life of IBD patients. Our findings provide a basis for interventions and services aimed at promoting the overall mental health of IBD patients of different genders. These data provide clinicians with a useful primer on which patients should be routinely screened for symptoms of common mental disorders and encourage gastroenterologists to screen and treat these conditions by gender, which may improve patient outcomes.

## 2. Materials and Methods

### 2.1. Study Design and Patient Population

This is a national multicenter cross-sectional survey on the mental psychological state and quality of life of IBD patients in China conducted by the Psychology Club of the IBD Group of the Gastroenterology Society of the Chinese Medical Association and the Chinese Association for Mental Hygiene from September 2021 to May 2022 involving 66 gastroenterologists and 2478 IBD patients from 42 hospitals in 22 provinces (including autonomous regions and municipalities) in China. Uniformly trained investigators explained to eligible patients the purpose, methods, voluntary participation, and harmlessness of the results of the survey and obtained informed consent, during which session standardized speech techniques were used. In this study, IBD patients were investigated via an online survey platform system. The study was approved by the Institutional Review Board of Renmin Hospital of Wuhan University, and informed consent was obtained from all patients.

Inclusion criteria: (1) Patients with a confirmed diagnosis of IBD by endoscopy, radiology, and histopathology [14]; (2) Age ≥18 years old; (3) Patients who can understand and agree to be investigated (e.g., willing to accept the doctor’s management and psychological investigation, obey the management of medication compliance, etc.); (4) Complete clinical data. Exclusion criteria: (1) Pregnant or lactating women; (2) Lack of reading and writing skills and communication difficulties; (3) History of mental illness; (4) Combined with other diseases that seriously affect the quality of life and/or mental state of patients (other intestinal diseases, malignant tumors, etc.). The screening flow chart of the patients is shown in Appendix A.

### 2.2. Study Assessments and Data Collection

#### 2.2.1. General Information Collection

We collected and sorted data on the gender and general clinical characteristics of IBD patients, including age, first visit, disease activity, disease course, main clinical manifestations (diarrhea, abdominal pain, hematochezia, parenteral complications, etc.), drugs used for treatment (5-aminosalicylic acid, glucocorticoid and immunosuppressants, biological agents, and traditional Chinese medicine), and surgery. The disease course was defined as the process between the onset of the symptoms and being investigated this time. According to the “consensus opinion on the diagnosis and treatment of IBD” [14], the modified Mayo score was used to evaluate the disease activity of UC patients (≤2 as remission, >2 as active), and the CDAI score was used to assess the disease activity of CD patients (<150 as remission, ≥150 as active).

#### 2.2.2. Generalized Anxiety Disorder 7-Item Scale (GAD-7) and Patient Health Questionnaire-9 (PHQ-9)

GAD-7 and PHQ-9 are two widely used self-report screening tools for anxiety and depression. Both scales instruct participants to indicate how often they have been bothered by each symptom over the last two weeks using a four-point Likert scale ranging from 0 (Not at all) to 3 (Nearly every day). Possible scores on the PHQ-9 range from 0 to 27, and on the GAD-7 from 0 to 21, with higher scores indicating higher levels of depression and anxiety. Scale scores of 10 or greater are typically used to indicate the probable diagnostic status of each of these measures [15,16,17]. Therefore, 10 points were taken as the cut-off point in this study. The score ranges for anxiety and depressive symptom severity in the results of both GAD-7 and PHQ-9 are as follows: minimal symptoms (0–4), mild (5–9), moderate (10–15), and severe (15+) symptoms.

#### 2.2.3. Pittsburgh Sleep Quality Index (PSQI)

PSQI was used to assess sleep quality in the last month, which is a well-validated self-assessment method. There are seven dimensions to this scale, and the total score is between 0 and 21. Scores >5 indicate significant sleep disturbance [18,19]. A score of 0–5 indicates good sleep quality, 6–10 indicates medium sleep quality, 11–15 indicates poor sleep quality, and 16–21 indicates very poor sleep quality.

#### 2.2.4. Inflammatory Bowel Disease Quality-of-Life Questionnaire (IBD-Q)

IBD-Q has been widely used to evaluate the quality of life of IBD patients, including four dimensions of bowel symptoms, systemic symptoms, emotional ability, and social ability, with a total of 32 items. Each item was divided into seven grades representing 1–7 points, and the total score ranged from 32 to 224. The higher the total score, the better the quality of life of the patient, with scores <169 being indicative of a poor quality of life [20,21].

### 2.3. Statistical Analysis

Excel 2021 was used to input and sort out the survey data, and SPSS 26.0 and R 4.1.0 software were used for the statistical analysis of the data. Count data were expressed as frequency and percentage, and the chi-square test or Fisher’s exact test were used for comparisons between groups. The odds ratio (OR) was considered significant if the 95% confidence interval (CI) did not include 1. The measurement data were expressed as mean ± standard deviation ($\bar{x}$ ± *s*), and t-tests were used to compare the two groups. Indicators with *p* < 0.05 were further included in logistic multivariate regression analyses, and independent influencing factors were screened out to construct nomogram prediction models. The receiver operating characteristic (ROC) curve was drawn, and the prediction ability of the model was evaluated by the area under the ROC curve (AUC). The bootstrap method was used to resample the model 1000 times for internal verification. The consistency index (C-index) was used to evaluate the discrimination of the nomogram prediction model, and the calibration curve was drawn to evaluate the model calibration. Decision curve analysis (DCA) was used to evaluate the clinical utility. *p* < 0.05 was considered statistically significant.

## 3. Results

### 3.1. Clinical Characteristics of IBD Patients of Different Genders

2478 IBD patients were enrolled, including 1547 males (62.4%) and 931 females (37.6%). The average age of females was higher than that of males (*p* = 0.001). The proportion of females in the active stage was significantly higher than that of males (65.5% vs. 59.6%, *p* = 0.003), and the disease course of different genders was also different (*p* = 0.005). Females were more likely to have hematochezia than males (48.2% vs. 39.4%, *p* < 0.001), and males were more likely to have complications than females (9.4% vs. 6.3%, *p* = 0.007). The proportion of females receiving 5-aminosalicylic acid treatment was higher than that of males (68.3% vs. 57.9%, *p* < 0.001), and the proportion of males receiving immunosuppressants (16.0% vs. 12.0%, *p* = 0.006), biological agents (56.3% vs. 43.2%, *p* < 0.001), and surgical treatments (14.3% vs. 8.9%, *p* < 0.001) was higher than that of females. There were no differences in the first visit, diarrhea, abdominal pain, extraintestinal manifestations, and receiving glucocorticoid treatment between males and females (all *p* > 0.05). However, there were no statistically significant differences between UC patients and CD patients of different genders except for age (all *p* > 0.05), see Table 1.

### 3.2. Psychological Symptoms, Sleep Quality, and Quality of Life in IBD Patients of Different Genders

#### 3.2.1. Scores of Psychological Symptoms, Sleep Quality, and Quality of Life in IBD Patients of Different Genders

The female GAD-7 (IBD: *p* < 0.001; UC: *p* < 0.001; CD: *p* = 0.014), PHQ-9 score (IBD: *p* = 0.001; UC: *p* = 0.001; CD: *p* = 0.192) and PSQI (IBD: *p* = 0.001; UC: *p* = 0.032; CD: *p* = 0.014) scores were significantly higher than those of males. Males had higher IBD-Q scores than females (IBD: *p* = 0.001; UC: *p* = 0.088; CD: *p* = 0.063), among which emotional ability was the most significantly different (IBD: *p* < 0.001; UC: *p* = 0.007; CD: *p* = 0.011) (Figure 1A–H and Appendix A).

#### 3.2.2. Psychological Symptoms, Sleep Quality, and Quality of Life in IBD Patients of Different Genders

As is shown in Figure 1 and Table 2, the anxiety rate among females was significantly higher than among males (IBD: 30.5% vs. 22.4%, *p* < 0.001; UC: 32.4% vs. 25.1%, *p* = 0.003; CD: 26.8% vs. 19.9%, *p* = 0.013), and there were differences in the severity of anxiety in different genders (IBD: *p* < 0.001; UC: *p* < 0.001; CD: *p* = 0.050). The depression rate among females was higher than among males (IBD: 33.1% vs. 27.7%, *p* = 0.005; UC: 34.4% vs. 28.9%, *p* = 0.031; CD: 30.6% vs. 26.6%, *p* = 0.184), and there were differences in the severity of depression in different genders (IBD: *p* = 0.004; UC: *p* = 0.022; CD: *p* = 0.312). The sleep disturbance rate among females was slightly higher than among males (IBD: 63.2% vs. 58.4%, *p* = 0.018; UC: 63.4% vs. 58.1%, *p* = 0.047; CD: 62.7% vs. 58.6%, *p* = 0.210), and the proportion of females with a poor quality of life was higher than that of males (IBD: 41.8% vs. 35.2%, *p* = 0.001; UC: 45.1% vs. 39.8%, *p* = 0.049; CD: 35.4% vs. 30.8%, *p* = 0.141). A total of 146 male patients (9.4%) (Figure 1M) and 132 female patients (14.2%) (Figure 1N) exhibited anxiety, depression, sleep disturbances, and a poor quality of life, respectively.

### 3.3. Influencing Factors of Psychological Symptoms, Sleep Quality, and Quality of Life in IBD Patients of Different Genders

#### 3.3.1. Univariate Analysis of Influencing Factors of Psychological Symptoms, Sleep Quality and Quality of Life in IBD Patients of Different Genders

In females, anxiety was associated with the first visit, disease activity, hematochezia, and use of different drugs (immunosuppressants and biological agents) (all *p* < 0.05), while in males, it correlated with the first visit, disease activity, disease type, and hematochezia (all *p* < 0.05). In females, depression was correlated with disease activity, diarrhea, and hematochezia (all *p* < 0.05), while in males, it was correlated with the first visit, disease activity, and abdominal pain (all *p* < 0.05). In females, sleep disturbance was associated with disease activity (*p* = 0.017), while in males, it was associated with the first visit (*p* = 0.029). In females, quality of life was associated with the first visit, disease activity, disease type, diarrhea, hematochezia, abdominal pain, use of biological agents, and surgical treatment (all *p* < 0.05), while in males, it was associated with the first visit, disease activity, disease type, disease course, diarrhea, hematochezia, abdominal pain, and extraintestinal manifestation (all *p* < 0.05). In addition, there was a significant correlation between anxiety, depression, sleep disturbance, and poor quality of life in both males and females (all *p* < 0.001) (Appendix A).

#### 3.3.2. Multivariate Logistic Regression Analysis of Influencing Factors of Psychological Symptoms, Sleep Quality, and Quality of Life in IBD Patients of Different Genders

Among males, disease activity, first visit, abdominal pain, anxiety, depression, and sleep disturbances were independent predictors of quality of life (Figure 2A), and anxiety, depression, sleep disturbance, and poor quality of life interacted with each other (Appendix A). In females, disease activity, first visit, diarrhea, abdominal pain, and depression were independent predictors of quality of life (Figure 2B), and anxiety, depression, and sleep disturbances were interlinked (Appendix A).

### 3.4. Construction and Verification of Nomogram for Predicting Quality of Life in IBD Patients of Different Genders

Since there were more than two variables affecting the quality of life of IBD patients, we included independent influencing factors of the quality of life of IBD patients of different genders (Figure 2A,B) and constructed nomogram prediction models for males and females (Figure 2C,D). The AUC values of the male and female prediction models were 0.771 (95% CI: 0.7466–0.7952) and 0.770 (95% CI: 0.7391–0.7998) (Figure 3A,D), with a sensitivity of 77.4% and 76.9% and a specificity of 66.5% and 64.6%, respectively, indicating high predictive value. The bootstrap method was used to repeatedly sample 1000 times to verify the modeling effect of the nomogram. The C-indices of the male and female prediction models were calculated to be 0.769 and 0.767, respectively, indicating that the discrimination of the models was high. The calibration diagrams of the two models showed that the calibration curves fitted well with the ideal curve (Figure 3B,E), indicating that the accuracy of the models was high. The DCA showed that when the risk threshold was 0.1–0.8 in both models, the clinical benefit of the nomogram models based on risk factors was significantly higher than that of the models without risk factors (Figure 3C,F).

## 4. Discussion

In recent years, the prevalence of IBD has been increasing worldwide and is projected to reach 1% globally by 2030, creating a significant global burden [22]. Studies have shown that IBD patients are more likely than the general population to suffer from psychological problems, such as anxiety and depression [23]. In addition, negative emotions such as anxiety and depression and sleep disturbance are key risk factors affecting the quality of life and prognosis of IBD patients and can even affect the quality of life of their caregivers [24,25,26,27]. The quality of life and psychological problems of IBD patients have attracted attention, and the gender differences in IBD have also been studied in recent years. However, there are currently no large-sample epidemiological data on the psychological state and quality of life of IBD patients of different genders, and this paper aims to address this.

In this study, all IBD patients were divided into male and female groups, and the general clinical data of each group were sorted out and analyzed. The analysis found that the average age of females, the proportion of females who had active IBD, and the proportion of females with hematochezia symptoms were all higher than those of males, and the proportion of males who had complications and received immunosuppressants, biological agents, and surgical treatment were higher than those of females. The results indicate that the physical condition of males might be relatively severe, which was mainly reflected in three aspects. First, males were more likely to have complications, and past studies have shown that IBD complications present gender specificity [28,29], and more complications often mean more severe diseases. Second, immunosuppressives and biological agents are usually used in moderate or severe patients, and the proportion of males using these drugs was significantly higher, which is consistent with the results of another study [30]. Third, males had a higher proportion of surgical treatment than females, and surgical treatment is usually suitable for patients with severe disease or severe complications. For example, ileocolonic surgery was more common among males [31] and had a high risk of recurrence after surgery [32]. The findings also indicated that females might be more vulnerable to psychological distress, which is mainly reflected in three aspects. First, the proportion of females in the active stage of the disease was higher than that of males, and patients in the active stage were more likely to suffer from psychological problems, leading to a decline in quality of life [26]. Second, the proportion of gastrointestinal symptoms, such as abdominal pain, diarrhea, and hematochezia, in females was higher than in males, and the occurrence of gastrointestinal symptoms is related to psychological abnormalities such as anxiety or depression, which may be a result of the brain–gut interaction [33]. Third, females were less likely than males to use biologics, and the use of biologics can significantly improve the quality of life of IBD patients [34]. In addition, it was surprising that there were no statistically significant differences in these clinical characteristics between UC patients and CD patients of different genders, although the overall trends were consistent, likely due to the insufficient sample size. Smoking can be considered the most extensively investigated and replicated environmental factor in IBD [35]. In the general population, the prevalence of smoking is always higher in males than in females [35]. A cohort study from Switzerland showed that smoking rates were significantly higher among female than male patients with CD (42.8% vs. 35.8%, *p* = 0.025) and that both anxiety and depressive symptoms were significantly associated with active smoking [35]. In comparison, among UC patients, there were no significant differences in smoking rates between females and males (13.2% vs. 17%), and no gender differences in anxiety and depressive symptoms levels were observed between smoking and non-smoking UC patients [35]. Therefore, there is a guideline recommendation that smokers with IBD should be encouraged to stop [36].

The results of this study showed that the incidence of anxiety and sleep disturbances in male IBD patients was lower than in female patients, and the quality of life in males was better than that in females, which was consistent with the conclusions of previous studies [9,37,38,39,40]. A recent systematic review [9] showed that there was a gender difference in the incidence of psychological abnormalities in IBD patients, in which the incidence of anxiety in males and females was 22.8% and 33.8%, and the incidence of depression in males and females was 16.2% and 21.2%, respectively. A series of studies [38,39] confirmed that females were more likely to suffer from anxiety or depression, which is consistent with the results of this study. Compared with males, females might have more emotional problems due to factors such as physical and psychological tolerance [41]. Therefore, in daily clinical diagnosis and treatment, we should pay more attention to whether females have mental and psychological abnormalities and their severity and provide effective health education and psychological support according to the specific situation. It was reported that there was a significant difference between the total score of PSQI and the gender of IBD patients, with females having worse sleep quality [37], which is basically consistent with the findings of this study. Therefore, female IBD patients need to take relevant measures to improve their sleep quality. A Dutch study [40] found that IBD-Q scores were lower in females than in males and that IBD-Q scores declined over time in females while they remained relatively constant in males. This survey also showed that males had a better quality of life than females, especially when it came to emotional abilities. Additionally, in many health-related quality of life studies, females consistently had lower health-related quality of life scores than males [42,43]. For many years, the explanation has presumably been that females might be more willing than males to consult a doctor about their symptoms and to incorporate disease factors into health-related quality of life reports [43,44], so even though they may actually be healthy, females tend to report lower health-related quality of life scores than males [43,44].

This study constructed the first nomogram model for predicting the quality of life of IBD patients of different genders with good predictive performance and high calibration. The nomogram model quantifies the predicted risk values, and clinicians can obtain the probability value of the quality of life of IBD patients of different genders via a simple calculation. The model found that disease activity, first visit, gastrointestinal symptoms, and depression all affected the quality of life of male and female IBD patients. It is well known that higher disease activity is associated with poorer quality of life [26]. Patients who visit the clinic for the first time might initially find it difficult to accept the occurrence of the disease, resulting in a heavy psychological burden and affecting their quality of life. In addition, the number of gastrointestinal symptoms in patients was positively associated with the incidence of anxiety and depression symptoms and negatively associated with quality of life [45], which is also consistent with this study in which we found that females had more gastrointestinal symptoms and a worse quality of life.

As for the potential mechanism of gender difference in IBD, we found that it may be mainly caused by genetic polymorphism (differential gene expression between genders), sex hormones, immune response, environmental factors, and intestinal microecology (Proteus, Clostridium, etc.) [46]. Several susceptibility genes are known to be associated with gender-specific risk of IBD; for example, there are multiple IBD genetic susceptibility loci on the X chromosome, such as Toll-like receptor 8 [47,48,49]. Differences in the levels of sex hormones and their receptors may be an important reason for the gender-specific manifestations of IBD in patients, as sex hormones are involved in regulating functions such as the gut microbiome, mucosal immunity, and the intestinal epithelial barrier [50,51]. Abnormal immune responses are involved in the pathogenesis of IBD, and the difference in immune environment and intensity between males and females may be the underlying reason for the gender difference in IBD [52,53]. Differences in exposure to environmental factors (e.g., smoking, antibiotics, and dietary habits) may also play an important role in gender differences in IBD [35,54]. In addition, gut microbiota can affect the central nervous system of the brain through metabolite neurotoxicity and then participate in the psychological abnormalities of IBD patients, meaning that the composition of the gut microbiota is affected by gender characteristics [55,56].

This study is the first multicenter large-sample survey on the mental state and quality of life of IBD patients of different genders. The results suggest that medical staff should pay more attention to the gender differences in the mental health of IBD patients and carry out timely intervention and dynamic monitoring, which is of great significance for the health management of IBD patients. In addition, the prediction model constructed in this study is the first nomogram to predict the quality of life of IBD patients of different genders. This model will be used for individualized risk screening, making patient care recommendations, and providing references for treatment decisions. However, this paper also has certain limitations. First, there are limitations inherent to cross-sectional studies, such as undocumented confounding factors and the inability to infer causal relationships for the stated associations. Second, the nomogram prediction model in this study has not been externally validated, and its applicability needs to be verified on a larger dataset. Third, disease activity was higher in females than in males, a factor that could explain to some extent some of the observed psychological differences. Fourth, IBD is a dynamic process, such that what is present at diagnosis may be completely different after treatment. Fifth, these data on IBD were not compared with data from the general population, where symptoms of anxiety and depression or poorer sleep quality are usually more common in female as well (regardless of whether they have IBD).

## 5. Conclusions

In conclusion, female IBD patients are more likely to have different degrees of anxiety, depression, and sleep disturbance than male patients, and male patients have a better quality of life than female patients. IBD physicians should fully consider the influence of gender differences in the diagnosis and treatment of the disease and adjust treatment plans in a timely manner in order to attain the best clinical efficacy.

## Figures and Tables

**Figure 1 jcm-12-01791-f001:**
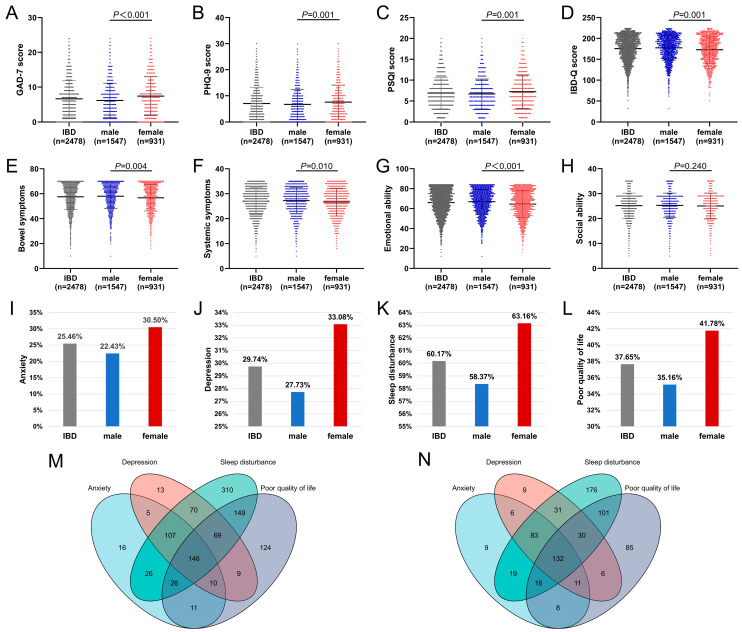
GAD-7 score (**A**), PHQ-9 score (**B**), PSQI score (**C**), IBD-Q score (**D**), anxiety rate (**I**), depression rate (**J**), sleep disturbance rate (**K**), and poor quality of life rate (**L**) of IBD patients. Venn diagram ((**M**): males; (**N**): females). IBD-Q score including 4 dimensions of bowel symptoms (**E**), systemic symptoms (**F**), emotional ability (**G**), and social ability (**H**). (IBD: inflammatory bowel disease; UC: ulcerative colitis; CD: Crohn’s disease).

**Figure 2 jcm-12-01791-f002:**
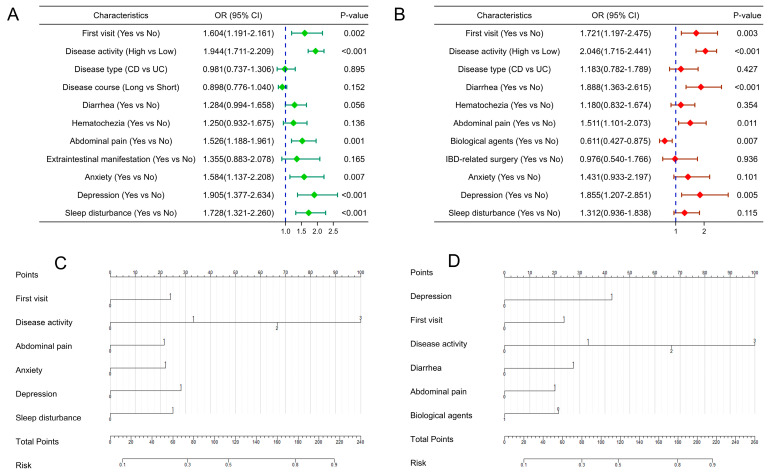
Forest plots (**A**,**B**) of independent influencing factors and establishment of nomogram models (**C**,**D**) for predicting quality of life in IBD patients of different genders ((**A**,**C**): males; (**B**,**D**): females). (Disease activity: 0 = Remission, 1 = Mildly active, 2 = Moderately active, 3 = Severely active. Other variables: 0 = No, 1 = Yes; IBD: inflammatory bowel disease; UC: ulcerative colitis; CD: Crohn’s disease; OR: odds ratio; CI: confidence interval).

**Figure 3 jcm-12-01791-f003:**
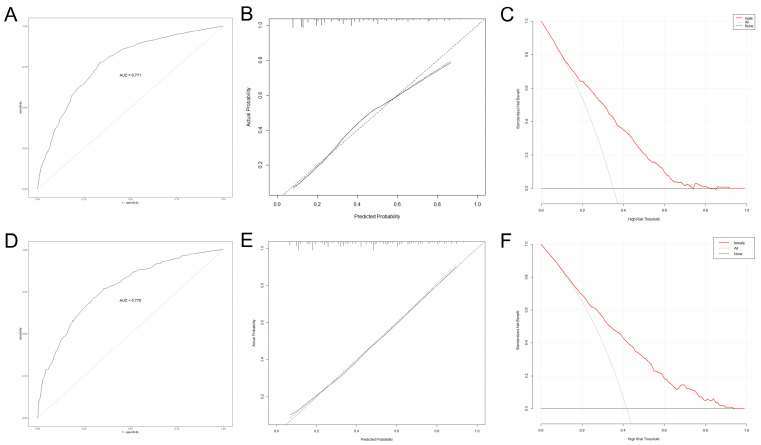
Internal validation and evaluation of nomogram models (the ROC curve (**A**), calibration curve (**B**), and decision curve (**C**) for males; the ROC curve (**D**), calibration curve (**E**), and decision curve (**F**) for females). (IBD: inflammatory bowel disease; UC: ulcerative colitis; CD: Crohn’s disease; ROC: receiver operating characteristic).

**Table 1 jcm-12-01791-t001:** Clinical characteristics of IBD patients of different genders.

Characteristics	IBD (*n* = 2478)		S	*p*-Value	UC (*n* = 1371)		S	*p*-Value	CD (*n* = 1107)		S	*p*-Value
	Male (*n* = 1547)	Female (*n* = 931)			Male (*n* = 754)	Female (*n* = 617)			Male (*n* = 793)	Female (*n* = 314)		
Age (years old, $\bar{x}$ ± *s*)	37.30 ± 12.71	39.06 ± 12.19	−3.378 *	0.001	42.22 ± 13.26	40.94 ± 11.69	1.899 *	0.058	32.63 ± 10.16	35.36 ± 12.33	−3.483 *	0.001
First visit (*n* (%))	297 (19.2)	196 (21.1)	1.254	0.263	194 (25.7)	149 (24.1)	0.452	0.502	103 (13.0)	47 (15.0)	0.752	0.386
Disease activity (*n* (%))			8.635	0.003			0.329	0.566			1.687	0.194
Remission	625 (40.4)	321 (34.5)			227 (30.1)	177 (28.7)			398 (50.2)	144 (45.9)		
Active	922 (59.6)	610 (65.5)	5.832	0.054	527 (69.9)	440 (71.3)	3.583	0.167	395 (49.8)	170 (54.1)	2.828	0.243
Mild	368 (39.9)	220 (36.1)			208 (39.5)	161 (36.6)			160 (40.5)	59 (34.7)		
Moderate	419 (45.4)	315 (51.6)			238 (45.2)	224 (50.9)			181 (45.8)	91 (53.5)		
Severe	135 (14.6)	75 (12.3)			81 (15.4)	55 (12.5)			54 (13.7)	20 (11.8)		
Disease course (*n* (%))			10.473	0.005			5.080	0.079			2.833	0.243
<2 years	560 (36.2)	379 (40.7)			303 (40.3)	280 (45.4)			257 (32.4)	99 (31.5)		
2~5 years	512 (33.1)	252 (27.1)			217 (28.9)	148 (24.0)			295 (37.2)	104 (33.1)		
>5 years	473 (30.6)	300 (32.2)			232 (30.9)	189 (30.6)			241 (30.4)	111 (35.4)		
Diarrhea (*n* (%))			2.631	0.105			0.019	0.891			0.570	0.450
Yes	928 (60.0)	589 (63.3)			511 (67.8)	416 (67.4)			417 (52.6)	173 (55.1)		
No	619 (40.0)	342 (36.7)			243 (32.2)	201 (32.6)			376 (47.4)	141 (44.9)		
Hematochezia (*n* (%))			18.653	<0.001			0.243	0.622			2.875	0.090
Yes	609 (39.4)	449 (48.2)			479 (63.5)	384 (62.2)			130 (16.4)	65 (20.7)		
No	938 (60.6)	482 (51.8)			275 (36.5)	233 (37.8)			663 (83.6)	249 (79.3)		
Abdominal pain (*n* (%))			0.991	0.319			2.769	0.096			2.461	0.117
Yes	874 (56.5)	545 (58.5)			368 (48.8)	329 (53.3)			506 (63.8)	216 (68.8)		
No	673 (43.5)	386 (41.5)			386 (51.2)	288 (46.7)			287 (36.2)	98 (31.2)		
Extraintestinal manifestation (*n* (%))			0.159	0.690			0.354	0.552			0.227	0.634
Yes	121 (7.8)	77 (8.3)			50 (6.6)	46 (7.5)			71 (9.0)	31 (9.9)		
No	1426 (92.2)	854 (91.7)			704 (93.4)	571 (92.5)			722 (91.0)	283 (90.1)		
Comorbidities (*n* (%))			7.362	0.007			0.330	0.565			1.243	0.265
Yes	146 (9.4)	59 (6.3)			16 (2.1)	16 (2.6)			130 (16.4)	43 (13.7)		
No	1401 (90.6)	872 (93.7)			738 (97.9)	601 (97.4)			663 (83.6)	271 (86.3)		
5-Aminosalicylic acid			26.931	<0.001			0.390	0.532			0.497	0.481
No	652 (42.1)	295 (31.7)			87 (11.5)	78 (12.6)			565 (71.2)	217 (69.1)		
Yes	895 (57.9)	636 (68.3)			667 (88.5)	539 (87.4)			228 (28.8)	97 (30.9)		
Mesalamine	881 (56.9)	631 (67.8)	28.647	<0.001	657 (87.1)	535 (86.7)	0.054	0.816	224 (28.2)	96 (30.6)	0.592	0.442
Sulfasalazine	42 (2.7)	24 (2.6)	0.042	0.837	36 (4.8)	23 (3.7)	0.903	0.342	6 (0.8)	1 (0.3)	0.167	0.683
Glucocorticoids			0.127	0.722			3.615	0.057			3.151	0.076
No	1311 (84.7)	784 (84.2)			595 (78.9)	512 (83.0)			716 (90.3)	272 (86.6)		
Yes	236 (15.3)	147 (15.8)			159 (21.1)	105 (17.0)			77 (9.7)	42 (13.4)		
Prednisone	147 (9.5)	88 (9.5)	0.002	0.967	88 (11.7)	54 (8.8)	3.114	0.078	59 (7.4)	34 (10.8)	3.355	0.067
Methylprednisolone	31 (2.0)	15 (1.6)	0.492	0.483	24 (3.2)	12 (1.9)	2.034	0.154	7 (0.9)	3 (1.0)	0.000	1.000
Dexamethasone	16 (1.0)	15 (1.6)	1.566	0.211	14 (1.9)	15 (2.4)	0.541	0.462	2 (0.3)	0 (0.0)	-	1.000
Immunosuppressants			7.493	0.006			1.429	0.232			2.473	0.116
No	1299 (84.0)	819 (88.0)			711 (94.3)	572 (92.7)			588 (74.1)	247 (78.7)		
Yes	248 (16.0)	112 (12.0)			43 (5.7)	45 (7.3)			205 (25.9)	67 (21.3)		
Thiopurines	225 (14.5)	95 (10.2)	9.736	0.002	31 (4.1)	30 (4.9)	0.450	0.502	194 (24.5)	65 (20.7)	1.778	0.182
Methotrexate	14 (0.9)	11 (1.2)	0.445	0.505	6 (0.8)	9 (1.5)	1.378	0.240	8 (1.0)	2 (0.6)	0.056	0.556
Biological agents			40.068	<0.001			3.125	0.077			1.645	0.200
No	676 (43.7)	529 (56.8)			531 (70.4)	461 (74.7)			145 (18.3)	68 (21.7)		
Yes	871 (56.3)	402 (43.2)			223 (29.6)	156 (25.3)			648 (81.7)	246 (78.3)		
Infliximab	599 (38.7)	255 (27.4)	33.035	<0.001	109 (14.5)	91 (14.7)	0.023	0.879	490 (61.8)	164 (52.2)	8.506	0.004
Vedolizumab	158 (10.2)	96 (10.3)	0.006	0.938	98 (13.0)	55 (8.9)	5.706	0.017	60 (7.6)	41 (13.1)	8.180	0.004
Adalimumab	71 (4.6)	32 (3.4)	1.937	0.164	17 (2.3)	10 (1.6)	0.706	0.401	54 (6.8)	22 (7.0)	0.014	0.907
Ustekinumab	48 (3.1)	21 (2.3)	1.541	0.214	2 (0.3)	1 (0.2)		1.000	46 (5.8)	20 (6.4)	0.130	0.719
Chinese herbs			1.422	0.233			0.109	0.742				0.487
No	1495 (96.6)	891 (95.7)			703 (93.2)	578 (93.7)			792 (99.9)	313 (99.7)		
Yes	52 (3.4)	40 (4.3)			51 (6.8)	39 (6.3)			1 (0.1)	1 (0.3)		
IBD-related surgery			15.576	<0.001			1.571	0.210			0.373	0.542
No	1326 (85.7)	848 (91.1)			739 (98.0)	610 (98.9)			587 (74.0)	238 (75.8)		
Yes	221 (14.3)	83 (8.9)			15 (2.0)	7 (1.1)			206 (26.0)	76 (24.2)		

Notes: S refers to statistical value; * refers to *t* value, others refer to χ^2^ value.

**Table 2 jcm-12-01791-t002:** Psychological symptoms, sleep quality, and quality of life in IBD patients of different genders (*n* (%)).

	IBD (*n* = 2478)		χ^2^	*p*-Value	UC (*n* = 1371)		χ^2^	*p*-Value	CD (*n* = 1107)		χ^2^	*p*-Value
	Male (*n* = 1547)	Female (*n* = 931)			Male (*n* = 754)	Female (*n* = 617)			Male (*n* = 793)	Female (*n* = 314)		
Anxiety			19.964	<0.001			9.017	0.003			6.138	0.013
No (GAD-7 < 10)	1200 (77.6)	647 (69.5)			565 (74.9)	417 (67.6)			635 (80.1)	230 (73.2)		
Yes (GAD-7 ≥ 10)	347 (22.4)	284 (30.5)			189 (25.1)	200 (32.4)			158 (19.9)	84 (26.8)		
Symptom Severity			31.594	<0.001			22.860	<0.001			7.821	0.050
Minimal	660 (42.7)	316 (33.9)			332 (44.0)	207 (33.5)			328 (41.4)	109 (34.7)		
Mild	540 (34.9)	331 (35.6)			233 (30.9)	210 (34.0)			307 (38.7)	121 (38.5)		
Moderate	232 (15.0)	165 (17.7)			131 (17.4)	115 (18.6)			101 (12.7)	50 (15.9)		
Severe	115 (7.4)	119 (12.8)			58 (7.7)	85 (13.8)			57 (7.2)	34 (10.8)		
Depression			7.966	0.005			4.677	0.031			1.765	0.184
No (PHQ-9 < 10)	1118 (72.3)	623 (66.9)			536 (71.1)	405 (65.6)			582 (73.4)	218 (69.4)		
Yes (PHQ-9 ≥ 10)	429 (27.7)	308 (33.1)			218 (28.9)	212 (34.4)			211 (26.6)	96 (30.6)		
Symptom Severity			13.296	0.004			9.612	0.022			3.572	0.312
Minimal	656 (42.4)	376 (40.4)			340 (45.1)	248 (40.2)			316 (39.8)	128 (40.8)		
Mild	462 (29.9)	247 (26.5)			196 (26.0)	157 (25.4)			266 (33.5)	90 (28.7)		
Moderate	260 (16.8)	162 (17.4)			134 (17.8)	109 (17.7)			126 (15.9)	53 (16.9)		
Severe	169 (10.9)	146 (15.7)			84 (11.1)	103 (16.7)			85 (10.7)	43 (13.7)		
Sleep disturbance			5.557	0.018			3.959	0.047			1.573	0.210
No (PSQI ≤ 5)	644 (41.6)	343 (36.8)			316 (41.9)	226 (36.6)			328 (41.4)	117 (37.3)		
Yes (PSQI > 5)	903 (58.4)	588 (63.2)			438 (58.1)	391 (63.4)			465 (58.6)	197 (62.7)		
Sleep quality			13.441	0.004			6.595	0.086			7.215	0.065
Good	644 (41.6)	343 (36.8)			316 (41.9)	226 (36.6)			328 (41.4)	117 (37.3)		
Medium	670 (43.3)	403 (43.3)			312 (41.4)	267 (43.3)			358 (45.1)	136 (43.3)		
Poor	202 (13.1)	151 (16.2)			112 (14.9)	103 (16.7)			90 (11.3)	48 (15.3)		
Very poor	31 (2.0)	34 (3.7)			14 (1.9)	21 (3.4)			17 (2.1)	13 (4.1)		
Quality of life			10.845	0.001			3.863	0.049			2.167	0.141
Not poor (IBD-Q ≥ 169)	1003 (64.8)	542 (58.2)			454 (60.2)	339 (54.9)			549 (69.2)	203 (64.6)		
Poor (IBD-Q < 169)	544 (35.2)	389 (41.8)			300 (39.8)	278 (45.1)			244 (30.8)	111 (35.4)		
IBD-Q score range			13.175	0.004			5.201	0.158			5.001	0.172
177–224	882 (57.0)	473 (50.8)			406 (53.8)	296 (48.0)			476 (60.0)	177 (56.4)		
129–176	556 (35.9)	364 (39.1)			275 (36.5)	251 (40.7)			281 (35.4)	113 (36.0)		
81–128	101 (6.5)	90 (9.7)			68 (9.0)	67 (10.9)			33 (4.2)	23 (7.3)		
32–80	8 (0.5)	4 (0.4)			5 (0.7)	3 (0.5)			3 (0.4)	1 (0.3)		

## Data Availability

The data that support the findings of this study are available from the corresponding author upon reasonable request.

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
