# Peer review of "Gender Differences in Psychological Symptoms and Quality of Life in Patients with Inflammatory Bowel Disease in China: A Multicenter Study"

_jcm, 2023, doi:10.3390/jcm12051791_

Round 1
Reviewer 1 Report
- - “7.69%”
Give a less precise figure
- - Use oxford comma in the whole text
- - Increase the size of Figure 1, Figure 2, Figure 3
- - “In terms of anxiety, females were associated with the first visit, disease activity, hem-196 atochezia, and the use of different drugs (immunosuppressants and biological agents) (all 197 P>0.05), while males were correlated with the first visit, disease activity, disease type, and 198 hematochezia (all P>0.05). In terms of depression, females were correlated with disease 199 activity, diarrhea, and hematochezia (all P>0.05), while males were correlated with the 200 first visit, disease activity, and abdominal pain (all P>0.05). …… In terms of quality of life, females were associated with the first visit, dis-203 ease activity, disease type, diarrhea, hematochezia, abdominal pain, use of biological 204 agents, and surgical treatment (all P>0.05), while males were associated with the first visit, 205 disease activity, disease type, disease course, diarrhea, hematochezia, abdominal pain, 206 and extraintestinal manifestation (all P>0.05).”
Associated or not associate? p < 0.05 or p >= 0.05?
- - It is obvious that the weight of the males is higher than that of females: why did you not use BMI?
- - What about difference in smoking habit in male and female?
Reviewer 2 Report
The article should be adapted to journal requirements (numerous editorial errors)
https://www.mdpi.com/journal/jcm/instructions
Authors should consider adding a few sentences to the introduction.
References are missing after the sentence on line 62.
Tables are not very readable.
The manuscript needs corrections to the English language.
The authors should add in the Methods section whether it was necessary to obtain the consent of the authors of the questionnaires and whether this consent was obtained.
Reviewer 3 Report
Manuscript ID: 2205740.
Gender differences in psychological symptoms and quality of life in patients with inflammatory bowel disease in China: a multicenter study.
Journal of Clinical Medicine.
This study by Liu C and colleagues explore the gender differences in patients with Inflammatory Bowel disease regarding anxiety, depression, quality of sleep and quality of life. The strenght of the study is the high number of patients included.
Nevertheless, there are some considerations the authors need to review.
Major comments:
- Line 37: as in other parts of the abstract before, consider write female data before male data.
- Line 90: if possible, specify in results (suplementary material) which diagnoses of mental illnes have patients excluded.
- Line 92 and 93: all the patients agreed to be investigated? There are no patients that refuse participate in the study?
- Line 96-100: it is necesary to explain better characteristics of IBD, specifically how disease activity was assessed and what means disease course.
-Line 116: reference number 14 is related to an interventional study about sleep in Inflammatory Bowel disease. A generic reference about PSQI index could be useful
. Example: Buysse DJ, Reynolds CF 3rd, Monk TH, Berman SR, Kupfer DJ. The Pittsburgh Sleep Quality Index: a new instrument for psychiatric practice and research. Psychiatry Res. (1989) 28:193–213. doi: 10.1016/0165-1781(89)90047-4
- Line 144-145: the average weight of a man, in general terms, is usually greater than that of a woman. Consider removing the weight data or replace it with the BMI. Also in line 261.
- Table 1: it will be useful to separate the contents in two tables. Another option is to repeat the heading ot the table in page 5. Similar situation in Table 2.
- Other limitations of the study:
- Disease activity is higher in female than male and this situation can explain some of the psicological diferences observed.
- Inflammatory Bowel disease is a dinamic process: situation at diagnosis can be totally diferent after treatment.
- These IBD-specific data should be contrasted with data of the general population, where anxious and depressive symptoms or poorer quality of sleep are also more frequent in women (regardless of whether there is IBD).
Minor comments:
- Line 54: ¨gender difference¨ is repeated, consider another different words.
- Line 85: consider delete All.
- Line 100: add ¨and¨ surgery.
- Line 171: more space between 1 and L.
- Line 209: a dot after () suplemmentary (as in line 217).
- Line 324: we have read a large 324 number of works of literature à consider delete this words.
Round 2
Reviewer 3 Report
It may be useful to know the number of patients who refuse to participate in the study (selection bias).
Author Response
Dear Reviewer 3,
First, we thank reviewer 3 for this positive and constructive suggestions.
We appreciate the reviewer’s attention to this critical comment. We have improved how many patients refused to be investigated in Supplementary Figure 1. Please see the latest version of Supplementary Figure 1 for details.
In all, I found the reviewer’s comments quite helpful, and I revised my paper accordingly. Thank you and the review again for your help!
Best regards,
Yours Sincerely